

# A genetic investigation in five Chinese families with keratoconus

Qinghong Lin[1,2,3,4,*], Xuejun Wang[1,2,3,*], Xiaoliao Peng[1,2,3],
Tian Han[1,2,3], Ling Sun[1,2,3], Xiaoyu Zhang[1,2,3] and Xingtao Zhou[1,2,3]

[1] Department of Ophthalmology, Eye and ENT Hospital of Fudan University, Shanghai, China
[2] NHC Key Laboratory of Myopia (Fudan University); Key Laboratory of Myopia, Chinese Academy of Medical Sciences, Shanghai, China
[3] Eye Institute and Department of Ophthalmology, Eye & ENT Hospital, Fudan University, Shanghai, China
[4] Refractive Surgery Department, Bright Eye Hospital, Fuzhou, China
* These authors contributed equally to this work.

Corresponding author
Xingtao Zhou,
Linqh19870624@163.com

## ABSTRACT

**Background:** This study investigated the genetic characteristics of five Chinese families with keratoconus (KC).

**Methods:** In the five families affected by KC, medical records, clinical observations, and blood samples were collected from all individuals. All KC family members ($n = 20$) underwent both whole exome sequencing of genomic DNA and Sanger sequencing to confirm the variants. Online software was utilized to analyze all variants, and the online server I-TASSER was employed for *in silico* predictions of the three-dimensional protein structures of the variants. The newly discovered variants and single nucleotide polymorphisms were further examined in 322 sporadic KC patients.

**Results:** The Pentacam tomographic composite index in those affected first-degree family members of the probands showed a pathological change. Five new variants were detected in the five probands and other affected members in their families: a heterozygous missense variant g.19043832C>T (p.Ser145Asn) in the homer scaffolding protein 3 (*HOMER3*) gene; a heterozygous missense variant g.99452113G>A (p.Gly483Arg) in the insulin-like growth factor 1 receptor (*IGF1R*) gene; a heterozygous missense variant g.55118280G>T (p.Trp843Leu) in the echinoderm microtubule-associated protein like 6 (*EML6*) gene; a heterozygous frameshift variant c. 1226_1227del (p.Gln410Glufs*17) in the DOP1 leucine zipper-like protein B (*DOP1B*) gene; and a heterozygous splice-site variant c.7776 +2T>A in the neurobeachin-like protein 2 (*NBEAL2*) gene. These variations were predicted to be potentially pathogenic and associated with KC.

**Conclusion:** Five novel variants in *HOMER3, IGF1R, EML6, DOP1B*, and *NBEAL2* genes were identified in this study and may be associated with the pathogenesis of KC. This study provides new information about the gene variants and their protein changes in KC patients. The findings should be explored further and could potentially be applied to the early diagnosis of KC before clinical onset.

## INTRODUCTION

Keratoconus (KC) is characterized by blurred vision caused by thinning and outward protrusion of the cornea. The etiology of KC includes genetic, behavioral, and environmental factors (*Lucas & Burdon, 2020*; *Nowak & Gajecka, 2011*; *Rong et al., 2017*). Among the KC cases, about 5–14% have a positive family history of KC (*Bykhovskaya, Margines & Rabinowitz, 2016*; *Loukovitis et al., 2018*). Autosomal dominant and recessive inheritance have been reported in KC with a family history (*Loukovitis et al., 2018*; *Nowak & Gajecka, 2011*). Bioinformatics studies have revealed the linkage and association of causative genes; however, only a few mutations have been identified in this disorder (*Bykhovskaya, Margines & Rabinowitz, 2016*; *Lin, Zheng & Shen, 2022*; *Lin et al., 2019*; *Loukovitis et al., 2018*).

In the last two decades, great advancements have been made in myopia correction surgery, and a large number of young myopia patients have come to the clinic. After a comprehensive preoperative ocular examination in these young patients, more corneal morphological changes indicating subclinical KC have been identified, triggering further exploration of KC, especially those resembling the subclinical stage of KC. In recent years, we studied two KC families and reported two novel variants associated with KC, *i.e.*, variants in collagen type V alpha 1 chain and transforming growth factor beta-induced (*TGFBI*) genes (*Lin, Zheng & Shen, 2022*; *Lin et al., 2019*). Our research has enriched the database (such as ClinVar) of variants associated with KC.

In this study, we tried to detect KC associated gene variants in five Chinese families with KC. Genes relevant to corneal development and the maintenance of function, such as homer scaffolding protein 3 (*HOMER3*), insulin-like growth factor receptor (*IGF-1R*), echinoderm microtubule-associated protein like 6 (EMAP like 6, *EML6*), DOP1 leucine zipper like protein B (*DOP1B*), and neurobeachin like 2 (*NBEAL2*), were selected for the study. Five novel variants in *HOMER3, IGF-1R, EML6, DOP1B*, and *NBEAL2* genes were identified.

## MATERIALS AND METHODS

### Participants and examinations

A total of 20 subjects from five Chinese families with KC and 322 sporadic KC subjects unrelated to the five families were enrolled in this study. Of the 20 individuals within two two-generation Chinese KC families, 12 were from families 1, 2, 3, and 4 (two patients and one unaffected member in each family) and eight were from family five (two patients and six unaffected members).

This study adhered to the diagnostic, staging, and grading criteria for KC as outlined in the Chinese Expert Consensus on the Diagnosis and Treatment for Keratoconus (*Corneal Disease Group in the Ophthalmology Association of the Chinese Medical Association, 2019*; *Lin et al., 2023*). All participants provided written informed consent. They received comprehensive ocular examinations comprising assessments of visual acuity, slit-lamp biomicroscopy, ophthalmoscopy, and evaluations of the cornea. Corneal examination was conducted using the Scheimpflug camera system (Pentacam; Oculus Optikgeräte GmbH,

Wetzlar, Germany). Approval for this study was obtained from the institutional review board of Fudan University (Approval No. 2022128; Shanghai, China), and all procedures were carried out in accordance with the principles of the Declaration of Helsinki.

## Variation identification and statistical analyses

Exome sequencing (ES) was conducted for all members of the five families using a methodology outlined in our previous study (*Lin, Zheng & Shen, 2022*). Leukocyte-derived DNA was extracted, followed by ES and enrichment of exonic sequences from the genomic DNA of KC subjects. Subsequently, data processing and analyses were performed. Reported variants (*e.g.*, in *TGFBI*, visual system homeobox 1, zinc finger protein 469 genes) and those occurring in KC subjects at frequencies ≤1% were assessed using the 1,000 Genomes Project. Next-generation sequencing (NGS) was utilized to assess the gene sequence and identifying variants. Only variants co-segregating with KC in affected family members, namely II:1 and I:1 in families 1 and 2, II:1 and I:2 in families 3 and 4, and II:1 and III:1 in family 5, were considered candidate variants. Polymerase chain reaction (PCR) and Sanger sequencing were utilized to confirm the NGS results of candidate variations in all family members, with PCR primers (Table S1) designed using Primer3 software (*Lin, Zheng & Shen, 2022*). Validation and analyses were carried out using the NCBI VARIANT, NCBI HomoloGene, and 1000 Genomes Project databases (*Lin, Zheng & Shen, 2022*). Whole genome sequencing (WGS) was performed in the sporadic KC patients (*Haarman et al., 2023*). WGS data of the sporadic KC patients were analyzed for any variant or SNP. The filtering conditions were set as frequencies <0.01, combined annotation-dependent depletion (CADD) value >10, and rare exome variant ensemble learner >0.5 to screen a variant as pathogenic or deleterious (*Lin, Zheng & Shen, 2022*). Analysis was carried out using the chi squared test ($x^2$ test) to explore the association of candidate variants or detected SNPs with susceptibility to KC. A significance level of $P < 0.05$ was considered statistically significant.

## Variant validation and cross-species conservation analysis

Variant validation and analyses were conducted following the methodology outlined in our previous work (*Lin, Zheng & Shen, 2022*). All variants underwent analysis using online software tools as detailed in our published report (*Lin et al., 2024*), with scoring in accordance with the American College of Medical Genetics and Genomics (ACMG) guidelines (*Richards et al., 2015*). The gnomAD database used was dbNSFP 3.3 (https://ngdc.cncb.ac.cn/databasecommons/database/id/930). In addition, the splice-site variation was also validated using the Human Splicing Finder (https://bio.tools/human_splicing_finder) and Splice AI (https://github.com/Illumina/SpliceAI). The conservation of the variant was predicted using computational toolkits (GERP++) (*Lin, Zheng & Shen, 2022*). Three-dimensional (3D) protein structures of the variants were generated utilizing the online server I-TASSER (https://zhanggroup.org/I-TASSER/). The effects of the variant on protein structure and function were explored using algorithms (Panther classification system) (*Lin, Zheng & Shen, 2022*).

## Cohort analysis

A total of 322 keratoconus patients were recruited for whole genome sequencing in PE150 mode of BGI DNBSEQ-T7 and variant calling using the BWA and GATK pipeline against the GRCh38 human genome (*Li & Durbin, 2009*; *McKenna et al., 2010*). The variants detected in the five genes, *i.e.*, *DOP1B* (chr21: 36156782–36294274), *EML6* (chr2: 54723499–54972025), *HOMER3* (chr19: 18929201–18941261), *IGF1R* (chr15: 98648539–98964530) and *NBEAL2* (chr3: 46979666–47009704) were extracted to calculate the minor allele count. Meanwhile, the data of minor allele count of each variant mentioned in above five genes were also archived from the ChinaMap (*Cao et al., 2020*) study, and were used as the control of the keratoconus cohort to perform chi-square test and ANNOVAR annotation (*Wang, Li & Hakonarson, 2010*). The exon position and -log10 (*P* value) of each variant were plotted using R software.

# RESULTS

## Clinical manifestations

The pedigrees are presented in Fig. 1. The clinical data of the KC patients are summarized in Table 1. In family 1, there were two subjects with KC. A 12-year-old boy in this family was the proband (II.1) (Figs. S1A, S1B). Another patient from family 1, subject I.1, the father of the proband, was 37 years old (Figs. S1C, S1D). There was no obvious abnormality in the cornea observed by slit-lamp biomicroscopy in the above two patients. Their clinical diagnosis was KC in the incipient stage. In family 2, the proband (II.1) was a 16-year-old male (Figs. S2A, S2B). The mother of the proband (I.1) was 50 years old and experienced central corneal thinning (Figs. S2C, S2D). In family 3, the proband (II.1) was a 20-year-old female with impaired vision in her left eye (Figs. S3A, S3B). Another patient (I.2) in family 3, the mother of the proband, was 46 years old (Fig. S3C). No significant abnormalities were observed using slit-lamp biomicroscopy. However, Belin analysis revealed suspicious Dp values in both eyes, and Belin/Ambrósio deviation (BAD-D) demonstrated pathological changes in the right eye (Fig. S3D). The proband in family 4 was a 32-year-old male with incipient KC (Fig. S4A). Belin analysis revealed suspicious Dp values in both eyes, and BAD-D demonstrated pathological changes in both eyes (Fig. S4B). The mother of the proband was 60 years old and experienced central corneal thinning (Fig. S4C). BAD-D also demonstrated pathological changes in both eyes (Fig. S4D). The proband in family 5 was a 27-year-old male (Fig. S5A). Belin analysis showed suspicious Dp values in both eyes, and BAD-D demonstrated pathological changes in both eyes (Fig. S5B). His mother (subject II.1) was 51 years old. Posterior corneal elevation (PCE) values at the thinnest point of the cornea were within the normal range; however, she suffered from central cornea thinning (Fig. S5C). No significant abnormalities were observed using slit-lamp biomicroscopy. However, Belin analysis revealed suspicious Dp values in both eyes and BAD-D demonstrated pathological changes in both eyes (Fig. S5D). Another patient (I.2) in this family was diagnosed as KC when he was alive.

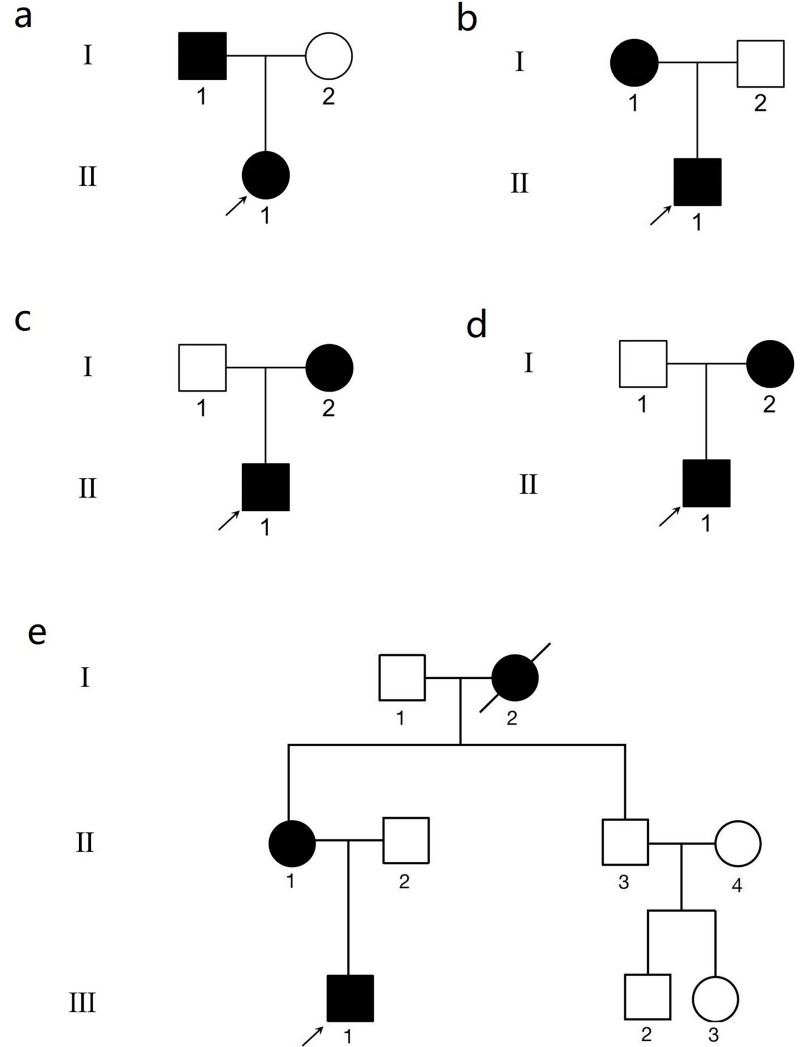

**Figure 1 The genogram of five Chinese family with keratoconus (KC).** The squares represent the males, and the circles represent the females. The solid symbols indicate the individuals with KC. The open symbols indicate the unaffected family members. (A) Family 1. Subject II.1 in family 1 was the proband (arrow). (B) Family 2. Subject II.1 in family 2 was the proband (arrow). (C) Family 3. Subject II.1 in family 3 was the proband (arrow). (D) Family 4. Subject II.1 in family 4 was the proband (arrow). (E) Family 5. Subject III.1 in family 5 was the proband (arrow).

## Detection and examination of the novel variant in the five families

Five previously unidentified variants were found in these five families and were absent in the healthy people. In family 1, a heterozygous *HOMER3* variation (g.19043832C>T, c.434G>A, p.Ser145Asn) was detected (Fig. 2A). It is located in exon 6. In family 2, a heterozygous missense variation in exon 6 of *IGF-1R* (g.99452113G>A, c.1447G>A, p.Gly483Arg) was detected (Fig. 2B). In family 3, a heterozygous missense variation (g.55118280G>T, c.2528G>T, p.Trp843Leu) in the *EML6* gene was detected (Fig. 2C). The variation is located in exon 17 of *EML6* gene. A heterozygous frameshift variation (c. 1226_1227del, p.Gln410Glufs*17) was identified in exon 10 of the *DOP1B* gene in

**Table 1  Clinical data of KC patients in the five families.**

| KC members | Sex | Age at diagnosis | CCT (μm) | Kmax | PCE (μm) | BSCVAs | Dp | BAD-D |
|---|---|---|---|---|---|---|---|---|
| **Family 1** | | | | | | | | |
| II.1 | M | 12 | 465 μm (OD), 423 μm (OS) | 49.6 D (OD), 66.7 D (OS) | 16 μm (OD), 62 μm (OS) | −2.00 DS/−2.50 DC × 170°→40/50 (OD), −3.00 DS/−3.00 DC × 10° →20/50 (OS) | 1.98 (OD) 9.78 (OS) | 3.25 (OD) 11.64(OS) |
| I.1 | M | 37 | 578 μm (OD), 577 μm (OS) | 44.0 D (OD), 44.8 D (OS) | 18 μm (OD), 14 μm (OS) | −2.00 DS/−1.00 DC × 175°→10/50 (OD), −2.50 DS/−0.50 DC × 15°→20/20 (OS) | 2.65 (OD) 2.56 (OS) | 2.88 (OD) 3.01 (OS) |
| **Family 2** | | | | | | | | |
| II.1 | M | 16 | 481 μm (OD), 436 μm (OS) | 42.5 D (OD), 61.6 D (OS) | 61 μm (OD), 67 μm (OS) | −3.00 DS/−1.00 DC × 180°→20/20 (OD), −2.50 DS/−6.00 DC × 5°→20/50 (OS) | 0.49 (OD) 10.02 (OS) | 0.82 (OD) 13.31 (OS) |
| I.1 | F | 50 | 474 μm (OD), 472 μm (OS) | 45.8 D (OD), 46.0 D (OS) | 2 μm (OD), 1 μm (OS) | −1.25 DS→20/20 (OD), −2.00 DS→20/20 (OS) | 1.28 (OD) 1.92 (OS) | 1.62 (OD) 1.55 (OS) |
| **Family 3** | | | | | | | | |
| II.1 | F | 20 | 525 μm (OD), 490 μm (OS) | 47.0 D (OD), 55.1 D (OS) | 8 μm (OD), 31 μm (OS) | +0.50 DS/−0.50 DC × 15°→20/20 (OD), −3.00 DS/−4.00 DC × 100°→30/50 (OS) | 2.36 (OD) 6.66 (OS) | 1.58 (OD) 6.61 (OS) |
| I.2 | F | 46 | 484 μm (OD), 475 μm (OS) | 45.6 D (OD), 45.5 D (OS) | 5 μm (OD), 4 μm (OS) | −1.50 DS/−1.00 DC × 110°→20/20 (OD), −0.50 DS/−1.25 DC × 80°→20/20 (OS) | 2.28 (OD) 1.99 (OS) | 2.09 (OD) 1.59 (OS) |
| **Family 4** | | | | | | | | |
| II.1 | M | 32 | 475 μm (OD), 477 μm (OS) | 45.2 D (OD), 44.8 D (OS) | 11 μm (OD), 13 μm (OS) | −5.00 DS/−1.00 DC × 180° →20/20 (OD), −1.50 DC × 5°→20/50 (OS) | 2.69 (OD) 3.04 (OS) | 2.26 (OD) 2.56 (OS) |
| I.2 | F | 60 | 466 μm (OD), 470 μm (OS) | 46.0 D (OD), 45.6 D (OS) | 15 μm (OD), 14 μm (OS) | −1.00 DS/−1.00 DC × 160° →20/20 (OD), −1.50 DS/−1.00 DC × 20° →20/20 (OS) | 1.37 (OD) 0.82 (OS) | 2.51 (OD) 2.50 (OS) |
| **Family 5** | | | | | | | | |
| III.1 | M | 27 | 500 μm (OD), 460 μm (OS) | 44.2 D (OD), 63.3 D (OS) | 13 μm (OD), 73 μm (OS) | −1.00 DS/−1.00 DC × 10°→20/ 20 (OD), −3.50 DS/−3.50 DC × 150°→20/50 (OS) | 3.02 (OD) 9.70 (OS) | 2.90 (OD) 9.88 (OS) |
| II.1 | F | 51 | 466 μm (OD), 469 μm (OS) | 46.6 D (OD), 45.9 D (OS) | 5 μm (OD), 6 μm (OS) | −1.50 DS/−0.50 DC × 170°→20/50 (OD), −0.75 DC × 165° →20/20 (OS) | 2.26 (OD) 2.32 (OS) | 2.17 (OD) 2.25 (OS) |
| I.2 | F | Passed away | | | | | | |

**Note:**
M, male; F, female; OD, right eye; OS, left eye; CCT, central corneal thickness; PCE, posterior corneal elevation; BSCVAs, best spectacle-corrected vision acuities; Dp, corneal thickness progression deviation; BAD-D, Pentacam tomographic composite index.

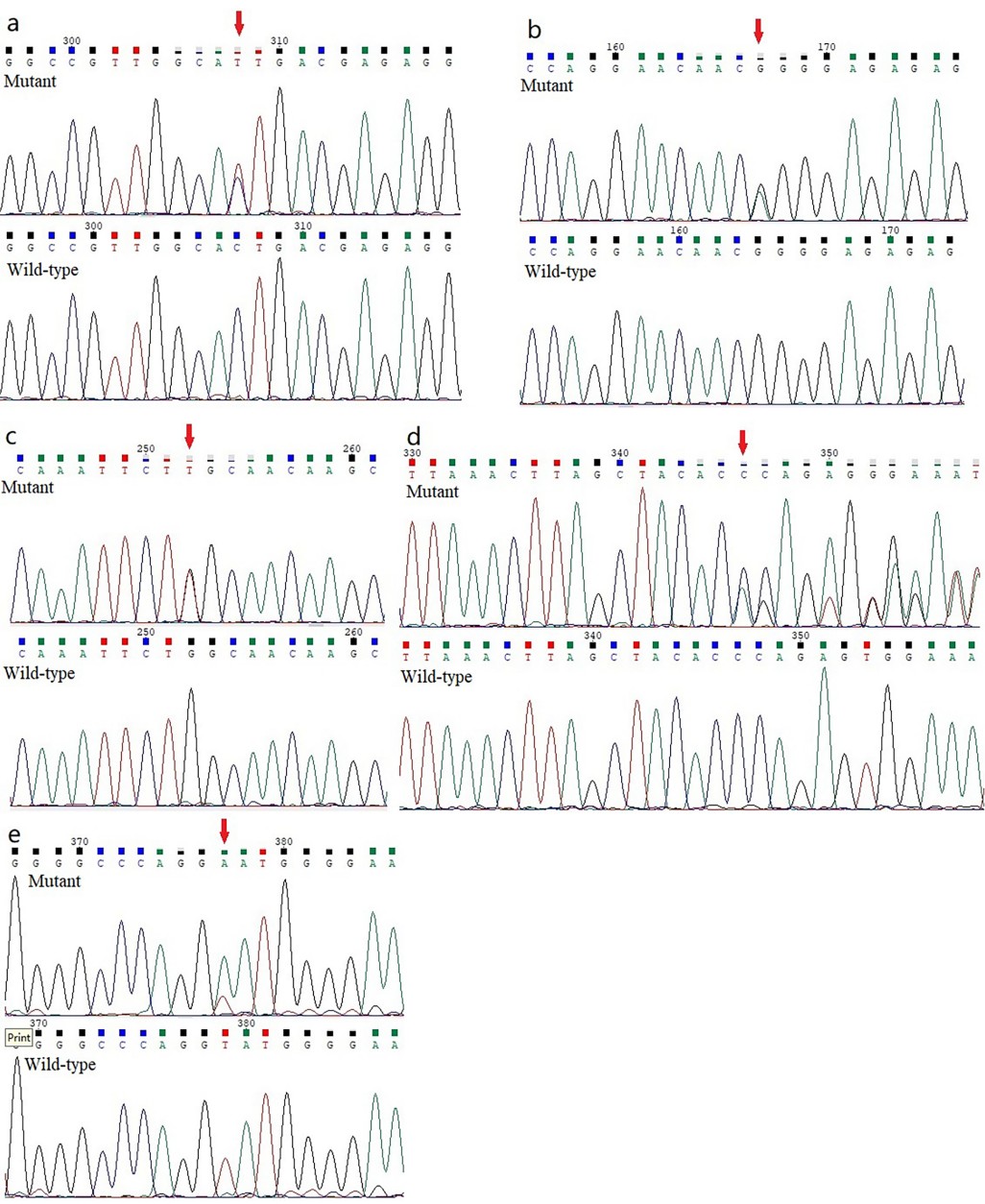

**Figure 2 Analysis report for the whole exome sequencing performed using the samples obtained from the KC members.** (A) A heterozygous *HOMER3* variant g.19043832C>T (c.434G>A) is shown (arrow). (B) A heterozygous nonsynonymous variant in *IGF-1R* (g.99452113G>A, c.1447G>A) is shown (arrow). (C) A heterozygous missense variation (g.55118280G>T, c.2528G>T) in the *EML6* gene is shown (arrow). (2D) The heterozygous frameshift variant c. 1226_1227del in *DOP1B* gene is shown (arrow). (E) A heterozygous splice-site variation c.7776+2T>A in *NBEAL2* gene is shown (arrow).

family 4 (Fig. 2D). Moreover, the heterozygous splice-site variation c.7776+2T>A in the boundary of exon 51-intron 52 in *NBEAL2* gene was identified in family 5 (Fig. 2E). These five variations were detected in all KC members from the five families and were absent in the unaffected members from the five families. All missense variations and frameshift

variations (in families 1–3 and 5) were predicted to be potentially harmful and showed high conservation levels according to software predictions (Table 2 and Figs. 3A–3C). The splice-site variation c.7776+2T>A is predicted to be pathogenic using the Human Splicing Finder and Splice AI. The prediction of the Human Splicing Finder showed that the result was B (the donor splice site was identified and the mutation was predicted to disrupt it). The result of the Splice AI also showed that the probability that the position 3:47049933 (=47049935-2) was used as a splice donor decreases by 0.98, and the variation probably induces the donor splice site loss. The conformational alteration of the proteins caused by these variations was revealed by 3D modeling and was compared with their wild-type proteins (HOMER3, IGF-1R, and EML6) (Figs. 4A–4C). Based on the ACMG guidelines, these five variations were predicted to be variants of uncertain significance (VUS) (Table 2).

**Linkage disequilibrium analysis of SNPs**

The five new variants were not detected in the sporadic KC patients and database of healthy people. However, 20 SNPs in *IGF1R*, 15 SNPs in *EML6* and five SNPs in *DOP1B* were observed with significantly higher frequencies in the sporadic KC patients than those in the healthy people ($P < 0.05$). Interestingly, strong linkage was detected between the following variants in *IGF1R, i.e.*, rs184761936 and rs1394192864, rs573663993 and rs138020194, rs138020194 and rs563075883, rs1431850000 and Chr15:98943439C>T, and rs1187528817 and Chr15:98955401C>CGTTT (Fig. S6A). Strong linkage was detected between the following variants in *EML6, i.e.*, rs173304 andrs354205 and rs354206 (Fig. S6B), rs1035896336 and Chr2:54938991C>A, and rs1371833489 and rs1295782150 (Fig. S6C). None of the variations in *HOMER3* (Fig. S6D) and *NBESL2* (Fig. S6E) showed significant linkage. Thus, these five genes may associated with keratoconus.

## DISCUSSION

KC is a multifaceted condition stemming from a combination of factors, including behavioral, environmental, and genetic influences. Several genes, including interleukin 1 beta, cadherin 11, negative regulator of ubiquitin-like protein 1, *collagen type XXVII alpha 1 chain*, and hepatocyte growth factor, have been linked to the familial manifestation of KC, with a handful of mutations identified (*Nowak & Gajecka, 2011*). Additionally, certain familial KC cases have associations with other genetic conditions such as Marfan's syndrome, atopy, and allergies (*Bykhovskaya, Margines & Rabinowitz, 2016*; *Lin, Zheng & Shen, 2022*; *Lin et al., 2019*; *Loukovitis et al., 2018*; *Nowak & Gajecka, 2011*; *Rong et al., 2017*). Exploring genetic characteristics across diverse KC families may provide insights into its underlying causes. In this study, five novel variants associated with familial KC were identified in *HOMER3, IGF-1R, EML6, DOP1B*, and *NBEAL2* genes. These genes were selected to study because of their function and role in corneal development and maintenance of function.

HOMER is a scaffolding protein and has been largely investigated in the nervous system. This protein has several splice variants and three paralogs (HOMER1, HOMER2, and HOMER3) (*Shiraishi-Yamaguchi & Furuichi, 2007*). *Català et al. (2021)* identified the

**Table 2 Computational predictions and ACMG classification of all identified variants and their frequency in the gnomAD genomes.**

| GENE | Variation | Protein change | Conservation analysis | Present in 1KG data (MAF (%)) | Present in gnomAD_genomes | Present in EVS data [MAF (%)] | SIFT prediction | Polyphen2 (score) | Mutation taster | Provean |
|---|---|---|---|---|---|---|---|---|---|---|
| HOMER3 (19p13.11; OMIM 604800) | c.434G>A (NM_004838.4) | Nonsynonymous (p.S145N) | HC | NO | NO | NO | Deleterious | Possibly damaging 0.799 | Disease causing | Neutral |
| IGF-1R (15q26.3; OMIM 147370) | c.1447G>A (NM_000875.5) | Nonsynonymous (p.G483R) | HC | NO | NO | NO | Deleterious | Probably damaging 1 | Disease causing | Deleterious |
| EML6 (2p16.1; OMIM 400954) | c.2528G>T (NM_001039753.2) | Nonsynonymous (p.W843L) | HC | NO | 0.00009692 | NO | Deleterious | Probably damaging 1 | Disease causing | Deleterious |
| NBEAL2 (3p21.31; OMIM 614169) | c.7776+2T>A (NM_001365116.2) | Splice-site variation | – | NO | NO | NO | – | – | Disease causing | – |
| DOP1B (21q22.12; OMIM 604803) | c. 1226_1227del (NM_001320714.2) | Frameshift Deletion (p.Gln410Glufs*17) | – | NO | NO | NO | – | – | Disease causing | Deleterious |

| GENE | Variation | Protein change | fathmm MKL (score) | grantham _distance | CADD score | GERP++ score | Clinical significance | ACMG |
|---|---|---|---|---|---|---|---|---|
| HOMER3 (19p13.11; OMIM 604800) | c.434G>A (NM_004838.4) | Nonsynonymous (p.S145N) | Deleterious (0.975) | 46 | 26 | 4.91 | Likely pathogenic | Uncertain significance (PM2_Supporting) |
| IGF-1R (15q26.3; OMIM 147370) | c.1447G>A (NM_000875.5) | Nonsynonymous (p.G483R) | Deleterious (0.976) | 125 | 34 | 5.54 | Likely pathogenic | Uncertain significance (PM2_Supporting+PP3_Moderate+PM3_Supporting) |
| EML6 (2p16.1; OMIM 400954) | c.2528G>T (NM_001039753.2) | Nonsynonymous (p.W843L) | Deleterious (0.991) | 61 | 34 | 5.92 | Likely pathogenic | Uncertain significance (PP3) |
| NBEAL2 (3p21.31; OMIM 614169) | c.7776+2T>A (NM_001365116.2) | Splice-site variation | Deleterious (0.931) | – | 24.8 | 5.04 | Likely pathogenic | Uncertain significance (PVS1_Moderate +PM2_Supporting+PM3_Supporting) |
| DOP1B (21q22.12; OMIM 604803) | c. 1226_1227del (NM_001320714.2) | Frameshift Deletion (p.Gln410Glufs*17) | – | – | – | – | Likely pathogenic | Uncertain significance (PVS1+PM2_Supporting) |

**Note:**

ACMG, American College of Medical Genetics and Genomics; HC, highly conserved; PM2, absent from controls (or at extremely low frequency if recessive) in Exome Sequencing Project, 1,000 Genomes Project, or Exome Aggregation Consortium; PP3, multiple lines of computational evidence support a deleterious effect on the gene or gene product (conservation, evolutionary, splicing impact, etc.); PM3, testing parental samples to determine whether the variant occurs in cis (the same copy of the gene) or in trans (different copies of the gene) can be important for assessing pathogenicity. For example, when two heterozygous variants are identified in a gene for a recessive disorder, if one variant is known to be pathogenic, then determining that the other variant is in trans can be considered moderate evidence for pathogenicity of the latter variant; PVS1, null variant (nonsense, frameshift, canonical +/−1 or 2 splice sites, initiation codon, single or multi-exon deletion) in a gene where loss of function is a known mechanism of disease. Variants with CADD score IIV 20 indicates the 1% most deleterious variants in human genome. Grantham matrix score predicts the effect of amino acid substitutions on the basis of their polarity and molecular volume; codon replacements can be categorized as conservative (score 51–100), moderately conservative (score 51–100), moderately radical (score 101–150), or radical (score ≥151).

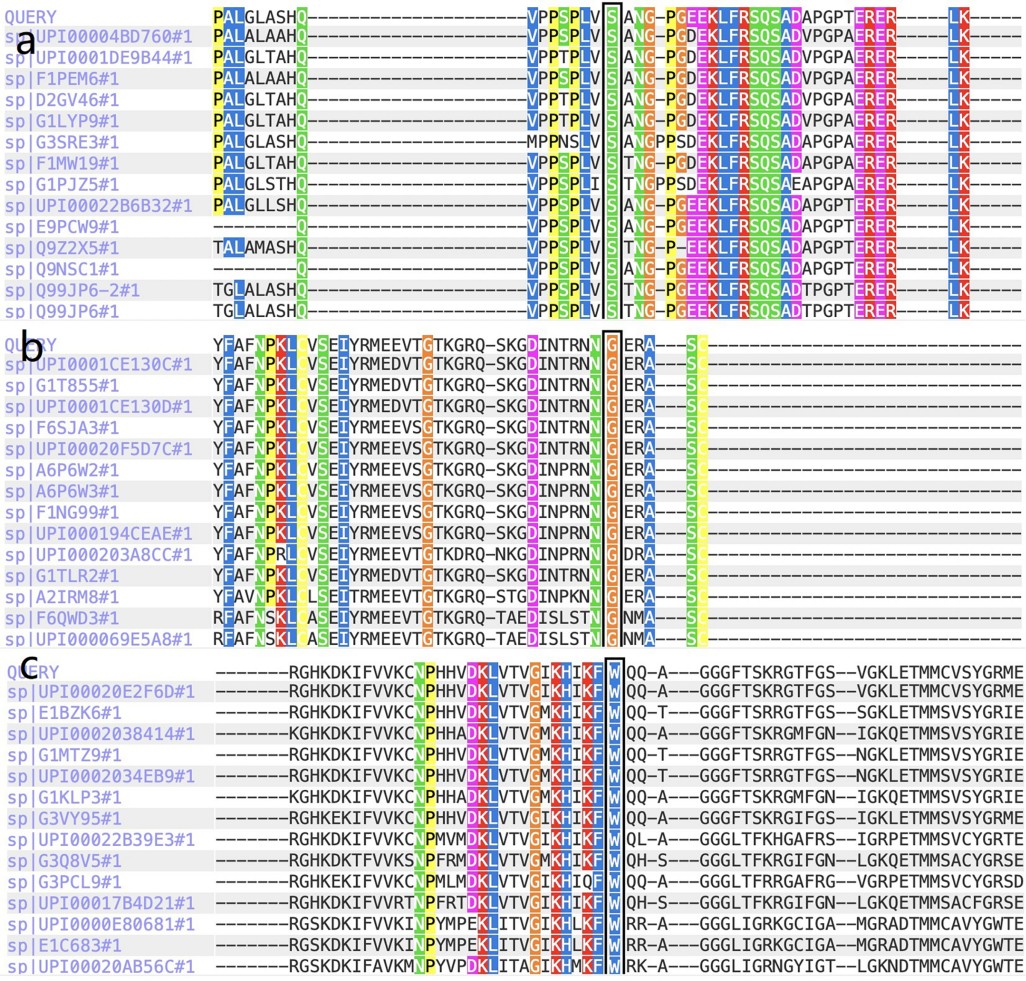

**Figure 3 The report for cross-species conservation analysis.** (A) The reports for cross-species conservation analysis of *HOMER3*. The 145 position of *HOMER3* is marked with a black box in the report. The multiple alignments indicate that the serine at codon 145 in *HOMER3* is highly conserved. (B) The report for cross-species conservation analysis of *IGF-1R*. The 483 position of *IGF-1R* is marked with a black box in the report. The multiple alignments indicate that the glycine at codon 483 in *IGF-1R* is highly conserved. (C) The report for cross-species conservation analysis of *EML6*. The 843 position of *EML6* is marked with a black box in the report. The multiple alignments indicate that the tryptophan at codon 843 in *EML6* is highly conserved.

expression of HOMER3 in the corneal epithelium and the limbus, indicating that it may have function in the cornea. The HOMER family plays an important role in cell function. HOMER3 is a cytosolic adaptor to involve in the regulation of G protein-coupled receptors (*Chiarello et al., 2013*). The presumed functions are anti-inflammation, antioxidation, reactive oxygen species homeostasis, and mitochondrial protection. Therefore, the redox status of cells and tissues are regulated with the involvement of HOMER, *e.g.*, cornea tissue. The imbalance of the HOMER involved activities is closely related to the development of KC (*Berbets et al., 2021*; *Tripathi, Khan & Chaudhury, 2022*; *Yapislar et al., 2022*). Several studies also have shown that HOMER3 can upregulate the store-operated calcium entry, mediating intracellular $Ca^{2+}$ concentration (*Jardin et al., 2012*; *Lang et al., 2013*).

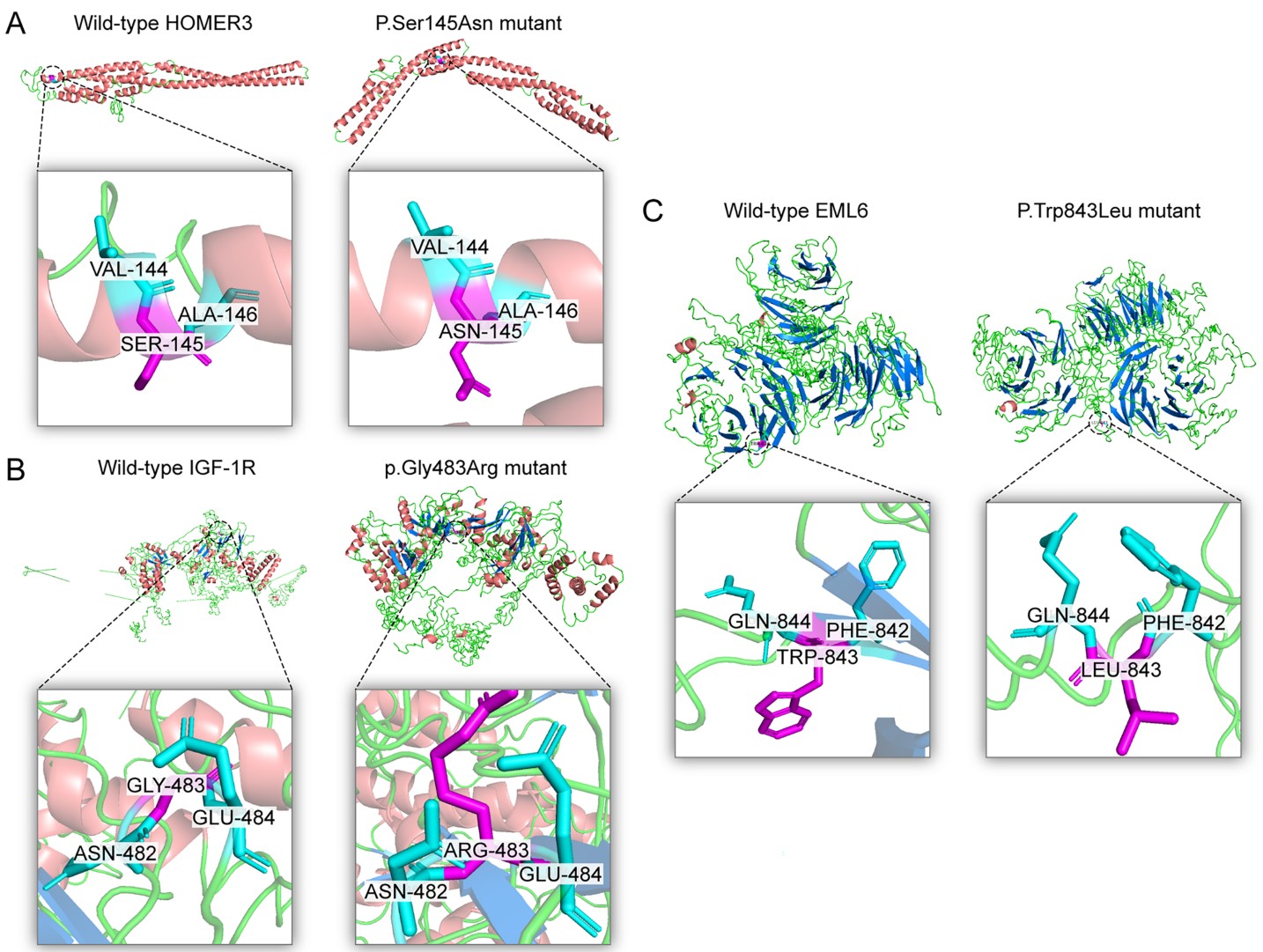

**Figure 4 Three dimensional structures of the proteins show the sites of variants.** The inset pictures are regional enlargement of the variants. (A) Three-dimensional (3D) modeling of wild-type *HOMER3* and p.Ser145Asn variant. (B) 3D modeling of wild-type *IGF-1R* and p.Gly483Arg variant. (C) 3D modeling of wild-type *EML6* and p.Trp843Leu variant.

Furthermore, wound repair of the corneal epithelium depends on $Ca^{2+}$-dependent signaling and cytoskeletal rearrangement, whereas corneal wound healing is retarded by hypercalcemia and cell viability is reduced (*Byun et al., 2016*; *Minns et al., 2016*; *Nagai et al., 2015*). In a similar way, treating band keratopathy by topical instillation of ethylenediaminetetraacetic acid (EDTA) is to chelate $Ca^{2+}$ and remove them from the cornea (*Al-Hity, Ramaesh & Lockington, 2018*; *Najjar et al., 2004*). Thus, we speculated that a lack of HOMER3 may disrupt $Ca^{2+}$ homeostasis, impair mitochondrial protective effect, retard corneal wound repair, and finally contribute to the development and progression of KC. In this study, the heterozygous *HOMER3* variant g.19043832C>T (c.434G>A) was detected in family 1, which is located in exon 6, causing a p.S145N amino acid change, and is anticipated to exhibit high conservation across species and potential harm according to

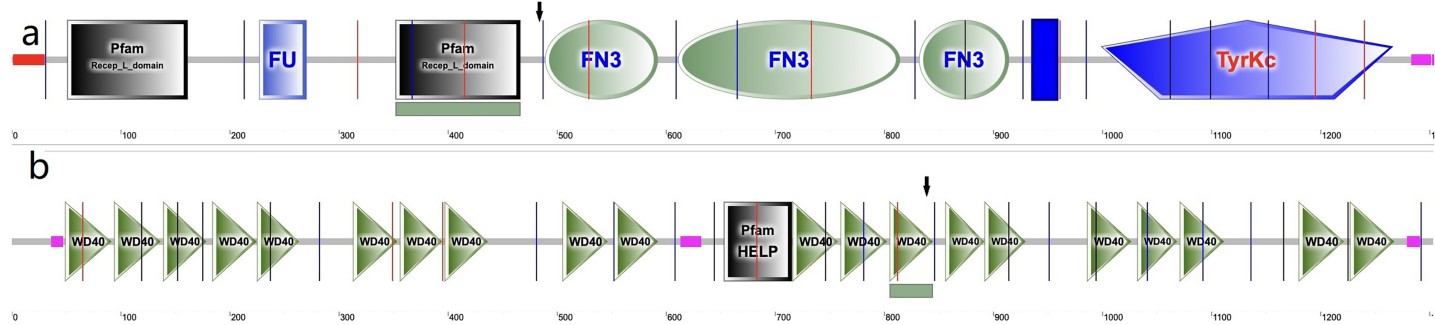

**Figure 5 Amino acid sequence schematic graphs show the variants in IGF-IR and EML6.** (A) The arrow shows the site of p.Gly483Arg variant in IGF-1R; (B) The arrow shows the site of p.Trp843Leu variant in EML6.

online software tools such as SIFT and Mutation Taster. Following the ACMG guidelines, it was deemed likely to be a VUS. Furthermore, its occurrence in the gnomAD population database is exceptionally rare (PM2). Assessment of the variant's conservation involved a GERP++ score evaluation, yielding a value of 4.91. To date, it is the first time to detect a *HOMER3* mutation in a KC family.

In the cornea, the IGF family plays a critical role in cell proliferation, differentiation, migration and wound healing, maintaining the cornea surface in a perfect status and ensuring a clear vision. The IGF family, IGF-1 and IGF-2, interacts with the hormone insulin (INSR) to make a functional system. IGF-1 has been confirmed to play a regulatory effect that promotes wound repair and maintains homeostasis in the cornea. For example, IGF-1 activates hybrid receptor and is followed by Akt phosphorylation to promote corneal epithelial cell proliferation (*Wu, Zhu & Robertson, 2012*). In addition, lamin-5 and β1-integrin can be upregulated by IGF-1 in the corneal epithelial cell migration through the phosphoinositide 3-kinase/Akt signaling (*Lee et al., 2006*). Moreover, IGF-1 is involved in the corneal stroma health through the effect on corneal keratocytes, which are the major type of cells in the corneal stroma. During corneal wound repair, IGF-1 promotes corneal wound repair through the upregulation of N-cadherin and downregulation of E-cadherin to enhance the adhesion of corneal fibroblasts and the epithelial-mesenchymal transition; whereas the normal function of the corneal fibroblasts could be blocked by picropodophyllin, an IGF-1 receptor (IGF-1R) inhibitor of IGF-1 (*Berthaut et al., 2011*). IGF-1R is a transmembrane protein with 1,367 amino acids and has extracellular and intracellular domains. It exists in the cornea and has greater affinity to IGF-1 than to IGF-2 and INSR (*LeRoith et al., 1995*; *Nakamura, Chikama & Nishida, 2000*). In the corneal endothelium, the expression of IGF-1R is about 25 times higher than insulin, keeping the balance of water in the corneal stroma through regulation of the Na/K-ATPase pump (*Feldman et al., 1993*). In this study, we identified a new variation in the *IGF1R* gene, which has not been previously reported in a KC family. In family 2, a heterozygous nonsynonymous mutation in exon 6 of IGF-1R (g.99452113G>A, c.1447G>A, p.Gly483Arg) was detected, which was absent in normal controls and all family members without KC. This variant is located in the middle of the Pfam domain (Recep_L_domain) and fibronectin type 3 domain (FN3), adjacent to the FN3 domain (489–592 aa) (Fig. 5A), which is anticipated to

exhibit high conservation across species and potential harm according to online prediction including provean, SIFT, Polyphen2, Mutation Taster, and fathmm MLK. Following the ACMG guidelines, it was deemed likely to be VUS. Furthermore, its occurrence in the gnomAD population database is exceptionally rare (PM2). The GERP++ score was 5.54, which suggests the variant has been conserved over time and is likely to have deleterious effects (PP3 and PM3).

EML6 is a large protein (1958 aa) in the microtubule-associated protein family. The effect of microtubules on protein traffic as well as the function of EML6 in the eye are not fully known (*Burute & Kapitein, 2019*). However, a study demonstrated an association between genome-wide association studies and refractive errors (especially astigmatism). KC patients usually have astigmatism, and this is also an important clinical indicator for monitoring the pathologic process (*Li et al., 2015*). In 2021, *Shinde et al. (2021)* revealed that EML6 can be detected in the corneal epithelium and stroma and identified two heterozygous missense variations in the Tryp-Asp (WD) dipeptide repeat domains. Furthermore, immunofluorescent staining of the corneal cytoskeletal structure in KC patients was performed in their study, which showed that EML6 may exist in the microtubules of the cytoskeletal network of cultured keratocytes. This is a very important finding, because EML1–4 but not EML5 and 6 are reported to be related to the regulation of microtubules. In this study, a missense mutation (g.55118280G>T, c.2528G>T, p.Trp843Leu) was detected and was predicted as probably damaging. The substitution p.W843L was in the WD dipeptide repeat domain (805–844aa) (Fig. 5B), a location that is considered highly conserved locus. The change in this domain is potentially pathogenic. Furthermore, the p.W843L variant is also adjacent to the boundary of two WD domains, which may have impact the atypical tandem β-propellar domain (*Shinde et al., 2021*). The consequence of it may alter the microtubule interaction. Following the ACMG guidelines, it was deemed likely to be a VUS; the GERP++ score was 5.92, which suggests that this variant has been conserved over time and is likely to have a deleterious effect (PP3).

The *DOP1B* gene is a member of the Dopey gene family and has been investigated in neuroscience because of its location on chromosome 21, the same region of Down syndrome, which is a 21 trisomy (*Rachidi et al., 2005*). *DOP1B* has been identified as a candidate gene for Peters anomaly (PA), and its expression can be detected in the ocular tissues of a PA patient such as the cornea, sclera, iris, lens, and retina (*Darbari et al., 2020*). *DOP1B* is involved in the process of cell development in the eye (*Dujon, 1996*; *Pascon & Miller, 2000*). In this study, a heterozygous frameshift variant c. 1226_1227del (p.Gln410Glufs*17) was identified in exon 10 of the *DOP1B* gene. This frameshift alteration took place within the coding region, involving the deletion of nucleotide CC (cytosine) at positions 1,226 and 1,227. This caused an amino acid change at residue 410, *i.e.*, glutamine changed to glutamic acid. Consequently, an early stop codon was induced after a frameshift of 17 amino acids, whereas a nonsense variation causes transcription issues, no products, or degradation. Thus, gene function is impaired. Per the ACMG guidelines, the variant is likely a VUS; this alteration represents a null variant in a gene where loss-of-function serves as compelling evidence of pathogenicity (PVS1). Furthermore, its occurrence in the gnomAD population database is exceedingly rare (PM2) and the GERP++ score was 5.04, which suggests that this

variant has been conserved over time and is also predicted to have a deleterious effect (PM3). Therefore, the frameshift variation c. 1226_1227del (p.Gln410Glufs*17) of the *DOP1B* gene was a pathogenic variation, which we reported as a novel variant found in a KC family.

*NBEAL2* encodes a cellular scaffold protein (2750 aa, 302 kDa), which is in the keratoconus library in NEIBank (neibank.nei.nih.gov). In 2021, *Shinde et al. (2021)* detected amino acid alterations in KC patients and a high level of NBEAL2 expression in the cornea. *NBEAL2* plays a role in cell metabolism and activity associated with lysosome-related organelles (*Cullinane, Schäffer & Huizing, 2013*; *Mayer et al., 2018*). While KC is a disorder with the abnormality in lysosome-related organelles (*Klintworth, 2009*). In this study, we identified a novel splice-site variation c.7776+2T>A in the exon 51-intron 52 boundary of *NBEAL2* gene in family 5. Per the ACMG guidelines, the variant was likely to be VUS. This alteration represents a null variant in a gene where loss-of-function serves as compelling evidence of pathogenicity (PVS1). Furthermore, its occurrence in the gnomAD population database is exceedingly rare (PM2). CADD score of above five variations was greater than 20, which also indicated a pathogenic effect.

In our study, we also investigated the association between the above five genes and the risk of KC in the Chinese population. Fourteen SNPs in these five genes were found to have a statistically significant higher frequency in the sporadic KC patients than the healthy controls. The data showed that the SNPs and variations in *HOMER3*, *IGF-1R*, *EML6*, *NBEAL2*, and *DOP1B* may be associated with an increased risk of KC.

Supported by the advanced analytic techniques of the eye, especially the anterior segment, investigation of the cornea in KC patients has been greatly developed. The genetic factors involved in the changes of corneal thickness, curvature, and PCE have been explored. In this study, parents of the probands in the five families and the proband in family 4 presented with early corneal changes, which were diagnosed as subclinical KC. These patients had thinner corneal thickness than normal controls or had increased PCE. The heredity of the disorder has been reported in previous studies, which compared the data of corneal topography between the first-degree relatives of a KC proband and healthy controls (*Belin, Jang & Borgstrom, 2022*; *Wang et al., 2000*). In this study, all participants received fully examinations and data analyses, especially corneal topography and Belin analysis. Through the analyses of the CCT, MASC, PCE, Dp, and BAD-D, we found that these indicators were abnormal in all relatives of KC probands who carried same gene variants. This indicates a potential association between a gene alteration and a corneal abnormality. In this study, the clinical characteristics of the immediate family members of probands was not typical of a KC diagnosis, and their visions were not much affected. This phenomenon demonstrated the importance of a thorough ocular examination, especially the cornea, to catch KC patients in the incipient stage (*Belin, Jang & Borgstrom, 2022*). Furthermore, with the application of advanced techniques of genetic analyses in clinic, ocular hereditary disorders such as KC can be detected and diagnosed very early before onset. This study provides novel information and enriches the database of genetic factors associated with KC.

In conclusion, five novel variants in *HOMER3, IGF-1R, EML6, DOP1B*, and *NBEAL2* genes were detected in this study. This is the first report that gene variants of *HOMER3*,

*IGF-1R*, and *NBEAL2* were identified in KC families. These five variations and associated genes may play a role in the pathogenesis of KC, from the *in silico* predictions and the population investigation. These discoveries broaden the range of recognized gene variants in KC, underscoring the need for additional studies.

## ACKNOWLEDGEMENTS

The authors would like to express their gratitude to all of the study participants for their cooperation. We thank Medjaden Inc. for their scientific editing of this manuscript.

### Funding

This study received funding support from the National Natural Science Foundation of China for Young Scholars (No. 82000929); the National Natural Science Foundation of China (No. 81770955); the Shanghai Sailing Program (No. 20YF1405000); the Project of Shanghai Science and Technology (No. 20410710100); the Clinical Research Plan of SHDC (No. SHDC2020CR1043B); the Project of Shanghai Xuhui District Science and Technology (No. 2020-015); the Project of Shanghai Xuhui District Science and Technology (No. XHLHGG202104); the Shanghai Engineering Research Center of Laser and Autostereoscopic 3D for Vision Care (No. 20DZ2255000); and the construction of a 3D digital intelligent prevention and control platform for the whole life cycle of highly myopic patients in the Yangtze River Delta (No. 21002411600). There was no additional external funding received for this study. The funders had no role in study design, data collection and analysis, decision to publish, or preparation of the manuscript.

### Grant Disclosures

The following grant information was disclosed by the authors:
National Natural Science Foundation of China for Young Scholars: 82000929.
National Natural Science Foundation of China: 81770955.
Shanghai Sailing Program: 20YF1405000.
Project of Shanghai Science and Technology: 20410710100.
Clinical Research Plan of SHDC: SHDC2020CR1043B.
Project of Shanghai Xuhui District Science and Technology: 2020-015.
Project of Shanghai Xuhui District Science and Technology: XHLHGG202104.
Shanghai Engineering Research Center of Laser and Autostereoscopic 3D for Vision Care: 20DZ2255000.
Yangtze River Delta: 21002411600.

### Competing Interests

The authors declare that they have no competing interests.

## Author Contributions

- Qinghong Lin conceived and designed the experiments, performed the experiments, analyzed the data, prepared figures and/or tables, authored or reviewed drafts of the article, and approved the final draft.
- Xuejun Wang performed the experiments, analyzed the data, prepared figures and/or tables, authored or reviewed drafts of the article, and approved the final draft.
- Xiaoliao Peng performed the experiments, prepared figures and/or tables, authored or reviewed drafts of the article, and approved the final draft.
- Tian Han performed the experiments, analyzed the data, prepared figures and/or tables, authored or reviewed drafts of the article, and approved the final draft.
- Ling Sun performed the experiments, authored or reviewed drafts of the article, and approved the final draft.
- Xiaoyu Zhang analyzed the data, authored or reviewed drafts of the article, and approved the final draft.
- Xingtao Zhou conceived and designed the experiments, authored or reviewed drafts of the article, and approved the final draft.

## Human Ethics

The following information was supplied relating to ethical approvals (*i.e.*, approving body and any reference numbers):

This study was approved by the institutional review board of Fudan University (Shanghai, China) (approval no. 2022128) and was performed in compliance with the Declaration of Helsinki.

## Data Availability

The variants are available in the NCBI ClinVar repository: SCV003842308, SCV003842309, SCV003842310, SCV003842311 and SCV003842314.

## Supplemental Information

Supplemental information for this article can be found online at http://dx.doi.org/10.7717/peerj.18037#supplemental-information.

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
