# Peer review of "A genetic investigation in five Chinese families with keratoconus"

_PeerJ, doi:10.7717/peerj.18037_

## Round 0.1 · original submission · Major Revisions

Thank you very much for your submission to PeerJ. Three reviewers have reviewed your manuscript. Please carefully look into the comments and suggestions from the reviewers and revise your manuscript accordingly.
Looking forward to your revised manuscript.

**Language Note:** The review process has identified that the English language must be improved. PeerJ can provide language editing services - please contact us at [email protected] for pricing (be sure to provide your manuscript number and title). Alternatively, you should make your own arrangements to improve the language quality and provide details in your response letter. – PeerJ Staff

Reviewer 1 ·

Basic reporting

The structure of this article conformed to the accepted format.

Experimental design

The submission should clearly define the research question, which must be relevant and meaningful.

Validity of the findings

Decisions are not made based on any subjective determination of impact, degree of advance, novelty or being of interest to only a niche audience.

Additional comments

Authors should revise the paper as follows:

Comments to the Authors:

- Abstract section: The authors should revise and harmonize all the variation reports to make them consistent with each other. For example: g.19043832C>T (p.Ser145Asn) in HOMER3 gene, and (g.99452113G>A, p.Gly483Arg) in IGF1R gene, among others. (Lines: 34-39)

- Abstract section: It is recommended to choose better keywords to improve the recognition of this research among keratoconus articles.
* * *
- Introduction section: Please rewrite the sentence “young myopia patients have come in great numbers to the clinic” (Line 57).
- Introduction section: Please revise and rewrite the paragraph for better understanding, (Lines 56-62).
- Introduction section: The authors mentioned “Our research enriched the database of the variants associated with KC” (Lines 62-63). Please briefly explain which database is being referred to?

- Introduction section: Why these genes ((HOMER3), (IGF-1R), (EMAP like 6, EML6), (DOP1B), (NBEAL2)) were selected?

- Introduction section: Provide an explanation of the selected genes and their role in keratoconus.
* * *
- Materials and methods section: What were the reasons for selecting this number of KC subjects and healthy controls for the study?
- Materials and methods section: What is the inclusion and exclusion criteria for recruiting families and controls? Please completely explain with details (e.g. biochemical factors or other analysis)
- Materials and methods section: Revise and clarify this paragraph, “Only the mutations co-segregating….” (Lines 89-92).
- Materials and methods section: Authors should explain how the PCR and Sanger sequencing confirmed the candidate mutations?
Additionally, provide more details about the methods of genetic analysis.
* * *
- Results section: Tables 1 and 2 lack explanations for the abbreviations. Please insert the full names of the abbreviations below the tables (e.g., OD for right eye, OS for left eye, M for male, BAD ……). Please provide the full name first and then use abbreviations.
- Results section, there is no demographic table?
- Results section: The description of the most important and relevant findings related to the purpose of the research should be included. Additionally, there is duplication of tables in the Results section.
* * *
- Discussion section: The discussion is too long and lacks a clear connection to the goal of the study. Revise and add proper references related to the claims made. Also, discuss other similar or contradictory research.

- Discussion section: The authors claim that “In this study, five novel mutations associated with familial KC were identified including HOMER3, IGF-1R, EML6, DOP1B, and NBEAL2 genes” (Lines 225-226).
However, no evidence is provided to directly link these genes to the susceptibility of KC.


- Discussion section: There is no declaration about the pathogenic effect of mutations in the susceptibility of KC. Please add proper references related to the effect of mutations in these genes and the susceptibility of KC.


Kind regards

Annotated reviews are not available for download in order to protect the identity of reviewers who chose to remain anonymous.

·

Basic reporting

The structure of the article is correct, but the writing needs to be improved as some sentences are ambiguous. Literature references are sufficient to the context provided, however in some places references are missing or improperly located in the sentence (details below).
The article contains professional figures and tables. As Table 1 contains necessary characteristics of KC patients the Results section concerning clinical aspects could be shortened. Moreover, the clinical characteristic of unaffected family members and controls is missing.

Additional comments:
1) Abstract: Instead of using ‘past medical history’ use ‘medical history’.
2) Abstract: The sentence ‘The clinical manifestations in those affected first-degree family members of the probands were significant’ is too general.
3) Consider using the word 'novel' whenever you refer to revealed new variants.
4) Remove from keywords or discuss more broadly ‘early diagnose’ of KC. How do your results justify it?
5) Line 59: Consider using the words 'forme fruste' or 'subclinical' instead of 'incipient'.
6) Line 60: There is a missing dot at the end of the sentence.
7) Line 86: Dot is placed before brackets.
8) Lines 193-194: Did any patient was diagnosed with KC post-mortem?
9) Were there any new variants/variants previously recognized in KC patients that did not meet the criteria for familial segregation?
10) Line 219: Sentence is too general and it's not correct (disease cannot be caused by etiology, etiology is rather a feature of disease).
11) Line 220: Please list some genes instead of writing only ‘couple of genes’.
12) Line 222: I don't think that putting atopy and allergy in one sentence with genetic syndromes is correct. Now it sounds like the familial form of keratoconus is related to familial allergy.
13) Line 224-226: This sentence is unclear. What does ‘KC subset’ mean? Please rephrase the sentence to state that variants were identified within genes.
14) Line 228: HOMER1, HOMER2, and HOMER3 are paralogs not subtypes. Also, genes’ names should be in italics.
15) Line 229: Please be more specific. What does ‘corneal disease’ mean?
16) Line 235-237: Be more specific in these two sentences.
17) Line 224: ‘Deficiency’ is not a good word as no quantitative experiments were performed. The reference is missing.
18) Line 249-250: Sentence is not grammatically correct.
19) Line 252: What does ‘perfect status’ mean?
20) Line 282: ‘In 2021, Shinde et al. (Fry et al., 2016) revealed that EML6 exists in the corneal epithelium and stroma and identified two heterozygous missense variations in the Tryp-Asp (WD) dipeptide repeat domains.’
What does ‘EML6 exist’ mean? Also, the reference is a review paper from 2016, whereas the sentence starts with the date 2021.
21) Line 284: Please rephrase ‘immunocytochemical feature’.
22) Line 287-288: Please describe in more detail why the discovery is important. Why is microtubule regulation important?
23) Line 298: Please rephrase the sentence (expression of a gene can be detected not the gene itself).
24) Line 230: The reference is improperly located or another one is missing.
25) Line 311: What does ‘significant NBEAL2 expression in the cornea’ mean?
26) Line 316: Authors use a noun instead of an adjective ‘pathogenic’.
27) Line 320-322: The sentence is not clear.
28) Please, give criteria for diagnosis of subclinical KC and explain your definition because there are no general criteria (often forme fruste and subclinical KC are confused).
29) Line 326: Whenever you refer to your results, it is better to use the words ‘here’ or ‘our study’.
30) Line 327: Which ‘indicators of the corneal morphology’ do you mean? Could you list them?
31) Please explain the colors in Figure 3.

Experimental design

The research question is well-defined but not original. Still, the study is relevant for the field as keratoconus etiology remains nebulous.
All used techniques meet the standards, although some information in the Materials and Methods section is missing, and therefore study could not be replicated by another investigator. Details regarding blood collection and DNA extraction (about used reagents) are lacking. Also, there is no information regarding ranges of DNA input for ES, reagents used for NGS library preparation, and NGS sequencing coverage. Please show the primer sequence, PCR reagents, and conditions.
Why only the 1000 Genomes Project was used as a database?
I've noticed there are underaged individuals included in the study group, therefore I assume the consent form has missing details (concerning the signature of legal guardians or the form should be worded differently). Errors also occur in the numbering of subsequent points of the form.

Validity of the findings

It would be beneficial to add to the Discussion the paragraph considering the study's weaknesses and strengths.
The last sentence from the study conclusion (“This study provides new information about the gene variants and their phenotypes in KC patients. The knowledge could be potentially applied to the early diagnosis of a KC patient before clinical onset.”) is not sufficiently backed by the obtained results. Why variants recognized in a single KC family could be applied in early diagnosis of KC?

Reviewer 3 ·

Basic reporting

Largely, the language is clear and professional throughout. However, a few deviations have been outlined below:
The appropriate nomenclature is “variant” rather than mutation. Please update throughout.
Please be consistent with where a reference is placed, either before or after the full stop. The references in the introduction and discussion are before the full stop, while in the methods (lines 86 and 95) it’s after the full stop.
There’s a full stop missing on line 60, the line should read “KC. In recent years, we studied…”.
Please move the clarification of which family is linked to which pedigree from line 108-109 “(1a. family 1; 1b family 2… family 5)” to the figure description or directly on the figure itself.
There is no need to list out all of the clinical measures for every affected family member in the results section. This is tabulated in Table 1 and much more interpretable in that format. Perhaps in this section clarify the diagnosis and age (keratoconus or subclinical disease).
Literature references are provided, but they tend to be old, unless from the author’s own research group. For example, the refences Nowak and Gajecka 2011 and Rong et al. 2017 are used on line 49 and 55 as part of the introduction. However, a much more recent review of keratoconus genetics was published in 2020 (Lucas and Burdon) and might be more appropriate.
The article is structured appropriately and self-contained, and the raw data (in the form of sequencing chromatograms) for the specific variants discussed have been shared.

Experimental design

This paper is primary research within the aims and scope of the journal.
The research question is not explicitly stated, though this is not commonly articulated in the field. The implied aim was to identify causative genetic variants in families with keratoconus.
The investigation performed in this paper is not well articulated and the methods are not currently described with sufficient detail. There is no clarity around variant filtering and candidate variant selection for the families. There needs to be more detail on how the variants were filtered, how many segregated in each family and why the variants that are presented in the paper were considered the “best” hits. Given that none of these genes have previously been associated with keratoconus before, they should be reported as candidate genes/variants. Without functional data or replication, the authors are overstating their findings. These variants have also appeared to have been classified as “likely pathogenic” (including in ClinVar) without any detailed classification/assertion criteria. This should be described in detail. If following the ACGM-AMP guidelines (current gold standard) these variants would not meet the strict criteria and should not be considered the cause of disease in these families without substantial further evidence.

Validity of the findings

Not all underlying data has been provided. As outlined above, there needs to be further detail on the filtering process and candidate variant selection.
Furthermore, the interpretation of the results could be improved by the following:
Please add a column to Table 1 that allows the diagnosis (keratoconus or subclinical keratoconus) to added. Please also include the unaffected relatives for comparison. Additionally, at it is stated on line 77 “all participants … received comprehensive ocular examinations”, this table could be greatly improved by including the summary statistics for the 100 controls (ie, the median and standard deviation for each of the measures). This would be more easily added as a column if the table was transposed so that each individual had their information in a column and the measures were along the rows.
Please update Table 2 to include the chromosomal position in hg38/GRCh38 as an additional column. Additionally, please include the minor allele frequency from gnomAD or similar (or note that it’s absent if that’s the case). Please also update the “phenotype” column heading to “protein change” or something else more appropriate.
The authors have deposited the data in ClinVar, however, they have copied and pasted directly from the paper and therefore it includes the numbers for references, please update this to PMIDs so that people reading the ClinVar entry alone can access the references easily. Also, in the paper (Table 2), the variants are listed as either “bengin” or “uncertain significance” however, in ClinVar, all have been listed as “likely pathogenic”. Please update the ClinVar entry and detail the criteria used.

Additional comments

This paper overstates the findings. The variants that have been identified, and therefore the genes that have been reported, should only be considered candidate genes. Without substantial functional evidence or replication in additional families, these variants/genes can not be considered the cause of disease at this stage. The inclusion of the filtering methods and other variants identified/considered would substantially improve the interpretability and therefore the readers trust in the robustness of the study.

---

## Round 0.2 · Major Revisions

Dear Authors,

Thank you very much for your efforts in revising your manuscript. However, a number of reviewer reports still contain issues that need to be addressed before this work would be appropriate for publication at PeerJ.

Reviewer 1 ·

Basic reporting

Clear and unambiguous
The article includes sufficient introduction and background

Experimental design

The submission clearly define the research question

Validity of the findings

The data presented in the paper had statistically, and controlled.
The conclusions have appropriately stated.

Additional comments

The authors have addressed the revisions and the manuscript could potentially be accepted for publication.

·

Basic reporting

I can see that the authors put a lot of work into the article and the writing style has been improved, but still, I have some concerns.
The authors did not respond step-by-step to my 'additional comments' therefore it was hard to follow their corrections.

Data availability statement (“The datasets generated and analyzed during the current study are available in the NCBI ClinVar repository, ClinVar accession number: SCV003842308, SCV003842309, SCV003842310, SCV003842311 and SCV003842314.”) should be rephrased as variants reported in the ClinVar repository cannot be treated as a full dataset generated in the study.
Moreover, I think that sharing the raw data would be a great benefit for the scientific community.

Were there any variants previously reported in KC patients that did not meet the criteria for familial segregation?

Some references still have errors, that is, they are incorrectly cited misleading the reader (details below).

In the method section authors added the sentence: “Whole genome sequencing (WGS) was performed in the sporadic KC patients and healthy controls (Lin et al., 2022).”
This reference (Lin Q, Zheng L, Shen Z. A novel variant in TGFBI causes keratoconus in a two-generation Chinese family. Ophthalmic Genet. 2022 Apr;43(2):159-163.) seems to be incorrect as in the original article the whole exome sequencing was performed. The authors should be precise about what additional analysis was performed (exome sequencing or genome sequencing).

There is also a mistake in the discussion regarding HOMER. Authors wrote: “ The function and role of HOMER in corneal disease have also been studied. In 2021, Català et al. identified the expression of HOMER3 in the corneal epithelium and the limbus (Català et al., 2021).”
In the cited paper (Català et al., 2021) the ocular disease was an exclusion criterion (“The tissues used for scRNAseq had no history of ocular disease, chronic systemic disease”).

Authors instead of citing the original publication cited the review paper (“...whereas the normal function of the corneal fibroblasts could be blocked by picropodophyllin, an IGF-1 receptor (IGF-1R) inhibitor of IGF-1 (Stuard et al., 2020).”)
Original paper: Berthaut A, Mirshahi P, Benabbou N, et al.. Insulin growth factor promotes human corneal fibroblast network formation in vitro. Invest Ophthalmol Vis Sci. (2011) 52:7647–53. 10.1167/iovs.10-5625.

Authors wrote: ‘In 2021, Shinde et al. revealed that EML6 exists in the corneal epithelium and stroma and identified two heterozygous missense variations in the Tryp-Asp (WD) dipeptide repeat domains (Fry et al., 2016).’
What does ‘EML6 exist’ mean? Also, the reference is a review paper from 2016, whereas the sentence starts with the date 2021.

Abstract: The sentence ‘The clinical manifestations in those affected first-degree family members of the probands were significant’ is too general.

The authors wrote in the abstract: "All variations in this study were predicted to be pathogenic and associated with KC". This sentence should be rewritten as the authors did not show sufficient evidence for this statement.

The ethnicity of new individuals included in the study (sporadic KC and additional controls) is mentioned only once (in the Results section). The aspect of possible differences in particular variant frequencies between various ancestry groups is left out.

In the method section authors did not mention the gnomAD database. Moreover, other used databases have missing information concerning the date of access, version, or hyperlink.

In Table 1 and Table 4, the number of decimal places should be equalized.
It would be appropriate to include the number of patients/controls (n=..) in the tables’ descriptions.

In figures’ legends and tables’ descriptions the word ‘mutation’ should be changed to ‘variant’.

There is a missing expansion of the abbreviation ‘PTVs’ in Table 5.

There is ‘0S’ instead of ‘OS’ in Table 2.

There are repeated typos in the abbreviation of posterior corneal elevation.

The HOMER is a protein family (not a single protein) containing three members. HOMER1, HOMER2, and HOMER3 are paralogs, not subtypes. Also, genes’ names should be in italics.

Authors wrote: “Thus, HOMER3 deficiency may disrupt Ca2+ homeostasis, impair mitochondrial protective effect, retard corneal wound repair, and finally contribute to the development and progression of KC.”
‘Deficiency’ is not a good word as no quantitative experiments were performed. The reference is missing.

Authors wrote: “Furthermore, the immunocytochemical feature of the corneal cytoskeletal structure in KC patients was investigated in their study...”.
The authors should be more specific/adequate, in the referenced study immunofluorescent staining was performed.

Authors wrote: “DOP1B has been identified as a candidate gene for Peters anomaly (PA), and can be detected in the ocular tissues of a PA patient such as the cornea, sclera, iris, lens, and retina”.
Please rephrase the sentence (expression of a gene can be detected not the gene itself).

Authors wrote: ‘significant NBEAL2 expression in the cornea’. What does it mean? Maybe the authors meant high.

Experimental design

No comments.

Validity of the findings

The study conclusion: “This study shows the potential of using candidate gene variants as indicators in the early diagnosis of KC in the future.”
Still, this conclusion seems to be not justified.

Reviewer 3 ·

Basic reporting

The English isn't always clear, it's sometimes unambiguous.

Literature references come across as referencing their own group's previous publications, rather than simply providing sufficient information.

There isn't a clear hypothesis for the research stated.

Experimental design

The aims are not clear. 5 genes were investigated for changes, but there wasn't clear rationale as to why these genes were selected, and other genes weren't included.

The methods were not reported in enough detail for the results to be replicated (for example there are no details on WES capture, read alignment or method for variant calling).

The authors still overstate their findings and have submitted their findings online, though they have reined it in a bit since earlier versions. The variants are still online as likely pathogenic, rather than the VUS that they report in the manuscript. This is NOT appropriate; these variants can be considered candidate variants in candidate genes at best.

Validity of the findings

The authors overstate their findings and do not present a clear rationale for the focus of their work (for example the selected genes the filtering strategies in the family where are "subclinical KC").

The statistics are not appropriate - they have not corrected for multiple testing, and therefore are reporting significant association where this is not appropriate.

The literature referenced in the paper is heavily weighted towards their own research group's research.

---

## Round 0.3 · accepted · Accept

Dear Authors,

I have evaluated the second revision of your article. In my opinion, you have carefully followed the reviewers' suggestions and revised the manuscript accordingly. Therefore, I am accepting the article for publication in PeerJ